# Dioxygen compatible electron donor-acceptor catalytic system and its enabled aerobic oxygenation

Jialiang Wei[1,2], Junhong Meng[1], Caifang Zhang[1], Yameng Liu[1] & Ning Jiao [1,2,3] ✉

The photochemical properties of Electron Donor-Acceptor (EDA) complexes present exciting opportunities for synthetic chemistry. However, these strategies often require an inert atmosphere to maintain high efficiency. Herein, we develop an EDA complex photocatalytic system through rational design, which overcomes the oxygen-sensitive limitation of traditional EDA photocatalytic systems and enables aerobic oxygenation reactions through dioxygen activation. The mild oxidation system transfers electrons from the donor to the effective catalytic acceptor upon visible light irradiation, which are subsequently captured by molecular oxygen to form the superoxide radical ion, as demonstrated by the specific fluorescent probe, dihydroethidine (DHE). Furthermore, this visible-light mediated oxidative EDA protocol is successfully applied in the aerobic oxygenation of boronic acids. We believe that this photochemical dioxygen activation strategy enabled by EDA complex not only provides a practical approach to aerobic oxygenation but also promotes the design and application of EDA photocatalysis under ambient conditions.

Visible-light photocatalysis has emerged as an important strategy for energy transfer and redox process, which has significantly contributed to the development of synthetic chemistry[1,2]. Under the assistance of photocatalysts, photocatalytic reactions typically occur under mild reaction conditions and employ simple oxidants or reductants like oxygen, air, or amines to replace stoichiometric reagents[3]. Traditionally, the π-conjugated aromatic dyes and transition metal complex chromophores which can harness visible light energy for chemical reactions with high efficiency, have been widely used in photochemistry (Fig. 1a)[4]. In recent decades, electron donor-acceptor (EDA)[5] complex photochemistry has garnered increased attention, which did not require any additional photocatalyst for successful execution and provided new opportunities for synthetic chemistry (Fig. 1a)[6]. Upon excitation of the EDA complex, an intra-complex single electron transfer (SET) occurs from the donor (D) to the acceptor (A), resulting in the formation of reactive radicals[7,8]. This strategy has been further utilized in valuable transformations including borylation[9], amination[10], alkenylation[11], alkylation[12], acylation[13], annulation[14,15] and coupling[16] (Fig. 1a)[17-19].

In general, EDA complex photochemistry requires stoichiometric amounts of specifically tailored donor and acceptor substrates, limiting the overall scope of these transformations[20]. Recently, employing a catalytic approach where either the donor[21-25] or acceptor[26] is used catalytically to form EDA complex, has gained attention as a satisfactory strategy. Despite the fantastic developments of EDA complex photochemistry, compared to other photocatalysts, this strategy is often incompatible with dioxygen. As electron-rich donors are usually air-sensitive and the biradical dioxygen can disrupt the desired radical and SET processes, this approach must be performed under an inert gas atmosphere (Fig. 1b)[27-32].

Interestingly, König and coworkers reported a consecutive visible light photoinduced electron transfer (conPET) using an air-stable electron-deficient π-conjugated aromatic molecule PDI (Fig. 1c)[33]. The intermediate generated during photoexcitation is stabilized by electron-withdrawing groups and π-conjugation effects. Inspired by this chemistry and the generating e⁻ pool and hole moieties[34,35] we hypothesized that the discovery of an appropriate acceptor catalyst

[1]State Key Laboratory of Natural and Biomimetic Drugs, Peking University, 100191 Beijing, China. [2]Changping Laboratory, Yard 28, Science Park Road, Changping District, 102206 Beijing, China. [3]State Key Laboratory of Organometallic Chemistry Sciences, Chinese Academy of Sciences, Shanghai 200032, China. ✉e-mail: jiaoning@pku.edu.cn

**Fig. 1 | EDA complex photochemistry and dioxygen activation. a** Overview of photochemistry. **b** Traditional EDA complex photochemistry theory and the problem associated with. **c** Research inspiration. **d** Mechanism design.

capable of producing an air-stable e⁻ pool would offer us an opportunity to develop an EDA system compatible with dioxygen (Fig. 1c).

Furthermore, this air stable e⁻ pool could transfer its free electron to dioxygen to generate an active superoxide radical anion, and enable exciting EDA complex catalyzed oxygenation reactions (Fig. 1c). Over the past few decades, organic chemists have taken a keen interest in the use of environmentally friendly molecular oxygen as the preferred oxidant for oxygenation reactions[36–49]. Although there have been considerable advancements in dioxygen activation assisted by

photocatalysts[50–56], the dioxygen activation and oxygenation via an EDA photocatalysis remains unknown. Herein, we report an air-stable EDA complex system that employs commercially available nitroaromatic compounds as effective organocatalytic acceptors and tertiary amines as electron donors (Fig. 1d). By using the catalytic amount of electron acceptor, the resulting dinitrobenzene radical anion is stabilized by the nitro groups, effectively inhibiting unproductive back-electron transfer (BET)[57,58]. This protocol efficiently initiated the generation of highly reactive superoxide radical anion under visible light excitation, thereby facilitating the aerobic oxygenation of boronic acids and producing hydroxylation products (Fig. 1d). Moreover, to the best of our knowledge, this is one of the few catalytic acceptor systems in the reported EDA complex photochemistry.

## Results and discussion

Based on the hypothesis, we examined the feasibility and properties of this proposed system. Further examination demonstrated the formation of a ground state EDA complex by UV/Vis spectroscopy (Fig. 2). Equimolar amounts of pale yellow 1,2-dinitrobenzene and colorless tri-n-butylamine were combined in acetonitrile, resulting in a darker solution (Fig. 2a). Subsequent measurement of the UV-Vis spectrum indicated a significant red shift and increased absorption of visible light (Fig. 2b). Commercially available 1,2-dinitrobenzene might not be pure, leading to a darker solution that affects light absorption. To resolve this, conduct simple column chromatography to get a pure, pale-yellow reagent. The observed absorption of visible light corresponds to the charge-transfer absorption of the EDA complex. This phenomenon is consistent with the expected effects of intermolecular interactions. While ensuring a constant final concentration of the acceptor, as the proportion of the Donor increases, the absorbance of the entire system gradually increases, displaying an initially rapid and then slower rate of increase. Once the Donor and Acceptor reach an equimolar ratio, the absorbance no longer exhibits a significant increase (Fig. 2c).

It can be observed that the maximum increment in absorbance of the EDA complex occurs at 490 nm (Fig. 2d). Using this reference wavelength, a Job's Plot experiment was conducted to investigate the absorbance increment of the EDA complex at different ratios. Surprisingly, the optimal binding ratio between the Acceptor and Donor was found to be 4:1 (Fig. 2e), which deviated from the commonly reported 1:1 binding ratio observed in most Acceptor and Donor pairs.

Furthermore, the addition of water (600 μL) to the EDA complex solution resulted in a significant reduction in system absorption and weakened the previously observed red shift phenomenon (Fig. 2f). When considering the dilution effect caused by the water addition (as demonstrated by a slight red shift in visible absorption upon adding water to the acceptor solution due to increased solvent polarity in the π-π* transition of nitroaromatics) (Fig. 2g), it was speculated that water acted as both a strong hydrogen bond donor and acceptor, thereby disrupting the formation of the EDA complex. This observation also provided an explanation for why the addition of 600 μL of water completely suppressed the subsequent oxygenation reaction involving superoxide radical species.

To demonstrate the presence of superoxide radicals, a highly reactive form of reactive oxygen species (ROS), a validation experiment was conducted. To monitor the fluorescence signal, the ROS fluorescent probe 2′,7′-Dichlorodihydrofluorescein (DCFH)[59] was introduced into the system, and an incremental increase in its positive fluorescence signal was observed with prolonged illumination time (Fig. 3a). This observation strongly supports the generation of ROS by the current dioxygen activation system. To further support our hypothesis, we utilized the superoxide-specific fluorescent probe dihydroethidium (DHE) to monitor the superoxide species, as previously reported in the literature[60]. Upon illumination in the present protocol, we clearly detected a positive fluorescence signal from the

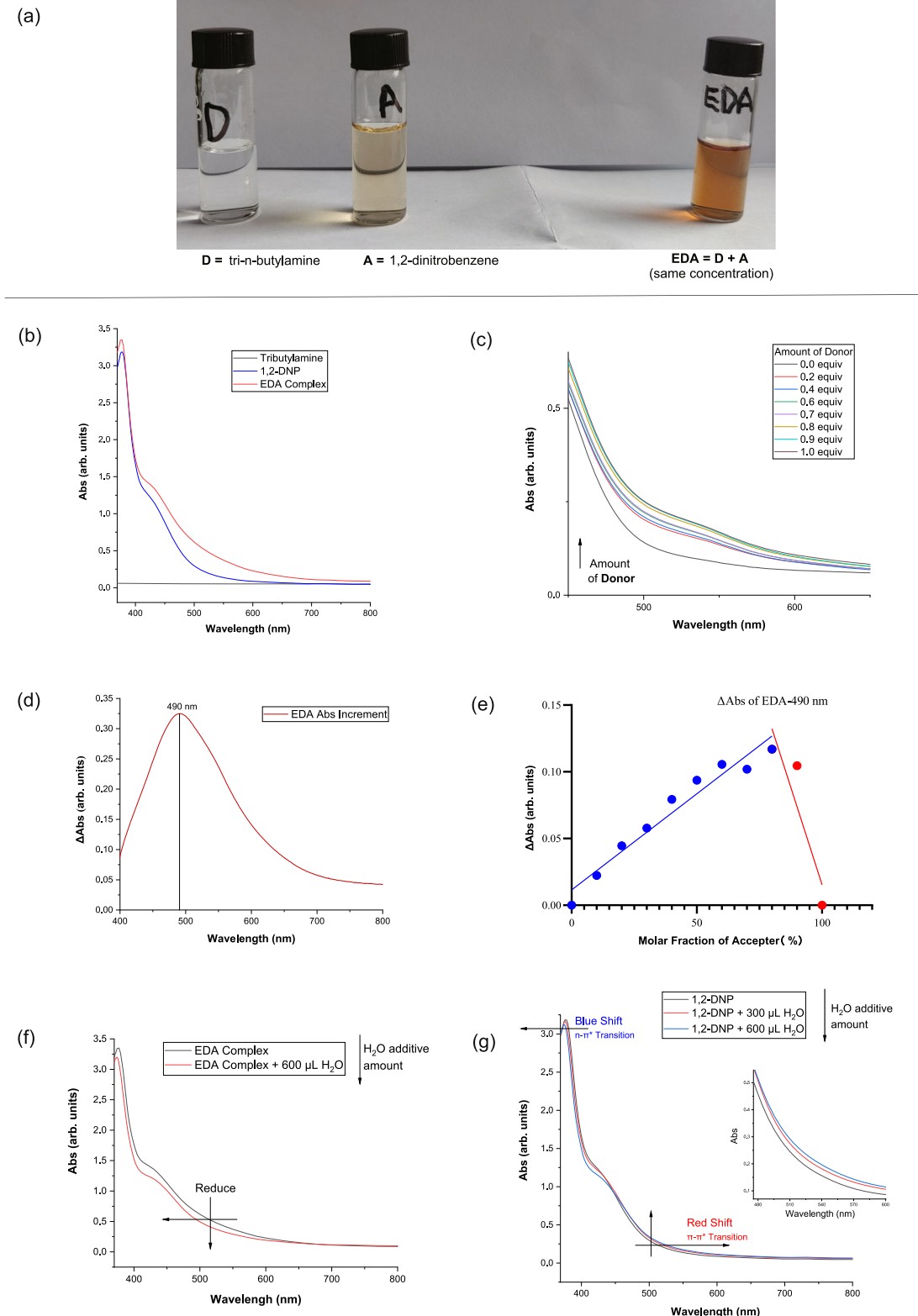

**Fig. 2 | Mechanistic experiments. a** Visual photograph of donor, acceptor and EDA complex. **b** Optical absorption spectra, the separate reaction components and appearance of the colored EDA complex. **c** UV-Vis absorption spectra of acceptor and that with different molar ratio of donor. **d** EDA Complex absorbance increment. **e** Job's Plot. **f** Water hydrolysis of the EDA Complex. **g** The impact of water on the UV-visible absorption spectra of acceptor. a.u. arbitrary units.

DHE probe (Fig. 3b), and the oxygenation product of DHE was successfully identified through mass spectrometry, providing conclusive evidence for the presence of superoxide species generated through dioxygen activation.

Having successfully demonstrated the compatibility and powerful capacity for the activation of oxygen to generate superoxide radical anions through this EDA complex photocatalytic system, we proceeded to employ organic boronic acid as substrates for the capture of

(a)

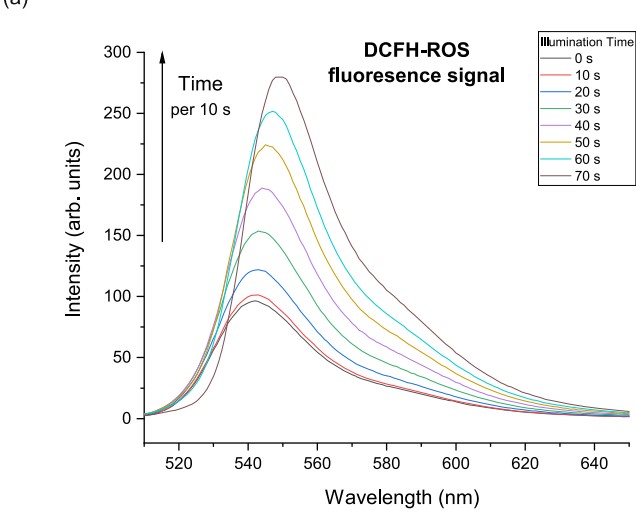

(b)

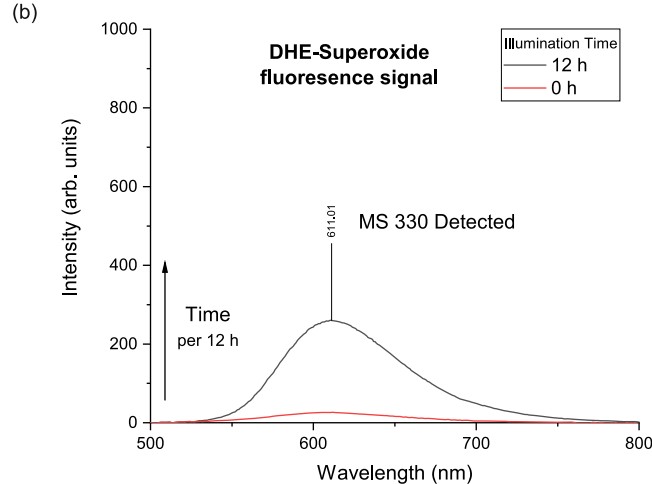

**Fig. 3 | Detection of ROS. a** ROS detection using DCFH fluorescence probe. **b** Superoxide detection using DHE fluorescence probe.

**Table 1 | Screening of reaction conditions.**[a]

| Entry | Changed conditions | Isolated yield |
|---|---|---|
| 1 | None | 99% |
| 2 | Toluene as solvent | Trace |
| 3 | DMSO as solvent | ND |
| 4 | MeOH as solvent | Trace |
| 5 | THF as solvent | Trace |
| 6 | DCM as solvent | Trace |
| 7 | DMF as solvent | 17% |
| 8 | EtOAc as solvent | Trace |
| 9 | 0.5 eq amine Donor | 56% |
| 10 | 1.5 eq amine Donor | 98% |
| 11 | 2.0 eq amine Donor | 97% |
| 12 | Tri-$n$-octylamine as Donor | 29% |
| 13 | DABCO as Donor | Trace |
| 14 | Et$_3$N as Donor | 28% |
| 15 | DIPEA as Donor | Trace |
| 16 | NPh$_3$ as Donor | Trace |
| 17 | 1,3-DNB as Acceptor | 34% |
| 18 | 1,4-DNB as Acceptor | 17% |
| 19 | No Light | ND |
| 20 | No Donor | ND |
| 21 | No Acceptor | ND |
| 22 | Under Ar | ND |
| 23 | Under Air | 69% |
| 24 | 600 uL H$_2$O as additive | Trace |

[a]Reaction conditions: 1a (0.5 mmol), amine donor (0.5 mmol), 1,2-DNB acceptor (0.025 mmol, 5 mol%) in solvent (4.0 mL) under blue LED irradiation and oxygen atmosphere (1 atm).

the generated superoxide intermediates. This pivotal step provided further compelling evidence of the operability and practical value inherent in the entire oxygen activation process of this strategy. In contrast to the reported stoichiometric peroxide oxidants[61–64], the EDA complex with 1,2-dinitrobenzene and tertiary amines was found very efficient to catalyzed the oxygenation process with O$_2$ as the oxidant and oxygen-source, yielding corresponding phenol product (Table 1). Acetonitrile was the most effective solvent for promoting the oxidation-oxygenation reaction, while other commonly used solvents gave unfavorable results (entries 2–8). Solvents with strong hydrogen bond donating properties, like methanol, could hinder the formation of the EDA complex and deactivate the desired catalytic process. We further identified 1,2-dinitrobenzene as the most effective acceptor, and tri-$n$-butylamine as the optimal donor to avoid the undesirable S$_N$Ar reaction (entries 12–18)[65,66]. Control experiments indicated that the presence of light, dioxygen, donor, and acceptor was essential for this dioxygen activation methodology, providing preliminary confirmation of the rationality and feasibility of the designed mechanism (entries 19–22). Under standard conditions, by simply changing the oxygen atmosphere to ambient air, the reaction proceeded smoothly and yielded the target product but with a little bit lower efficiency (entry 23). It is noteworthy that only catalytic amount of electron acceptor (5 mol%) was employed for this highly efficient aerobic oxygenation (Table 1).

After determining the optimized reaction conditions, we carried out experiments to evaluate the process of dioxygen activation using various boronic acid substrates (Fig. 4). The optimized conditions enabled us to achieve high yields of the oxygenation products. Notably, the reactivity of the dioxygen activation system displayed minimal sensitivity to the positional and electronic properties of the functional groups. Both electron-donating and electron-withdrawing groups were effectively accommodated by the system (Fig. 4). Additionally, functional groups that are typically easily oxidized by oxidants, such as alkynes (**2b**), sulfide groups (**2c**), alkenes (**2d**), aldehydes (**2f**), and benzyl position (**2y**), could be preserved under the optimized reaction conditions. These strong oxidant sensitive functional groups exhibited excellent compatibility within this oxidation-oxygenation system, allowing their preservation without any oxidative damage. This mild oxidation system demonstrated a wide range of substrates, making it promising for selectively oxidizing boryl groups in complex drug molecules or bioactive compounds containing multiple sensitive functional groups. The high tolerance of the reaction system towards functional groups suggested a gradual and suitable activation of oxygen, resulting in the generation of superoxide anion radicals during photocatalysis.

Moreover, besides conventional aromatic boronic compounds, alkyl-boronic acids also underwent efficient oxygenation reactions in this system, yielding the corresponding alkyl alcohol products. It is

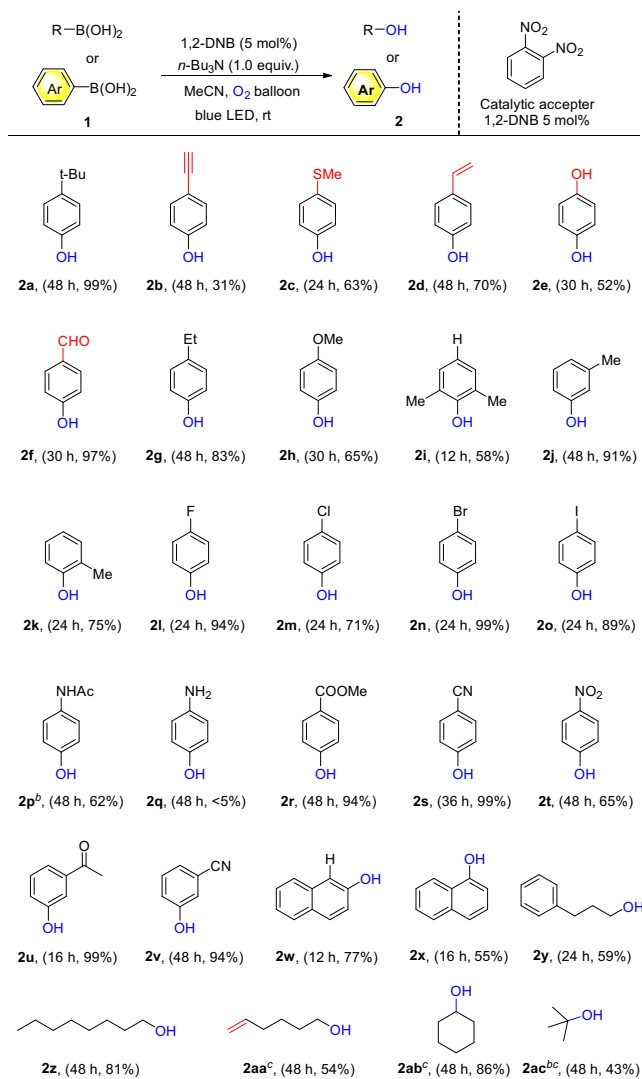

**Fig. 4 | Substrate scope[a].** [a] Conditions: Reactions were conducted on 0.5 mmol scale using 1.0 equiv. amine, 5 mol% 1,2-DNB at 0.125 M. Isolated yields after chromatographic purification. [b] boronic acid pinacol ester was used as starting compound. [c] H-NMR Yield in CD₃CN due to its low boiling point.

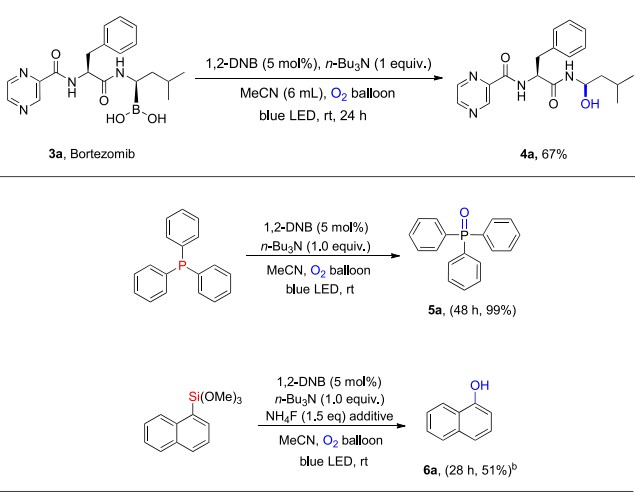

**Fig. 5 | Late-stage modification of drug molecule and diversifying the applications[a].** [a]Conditions: Reactions were conducted on 0.5 mmol scale using 1.0 equiv. amine, 5 mol% 1,2-DNB at 0.125 M. Isolated yields after chromatographic purification. [b]1 equivalent of NH₄F as additive.

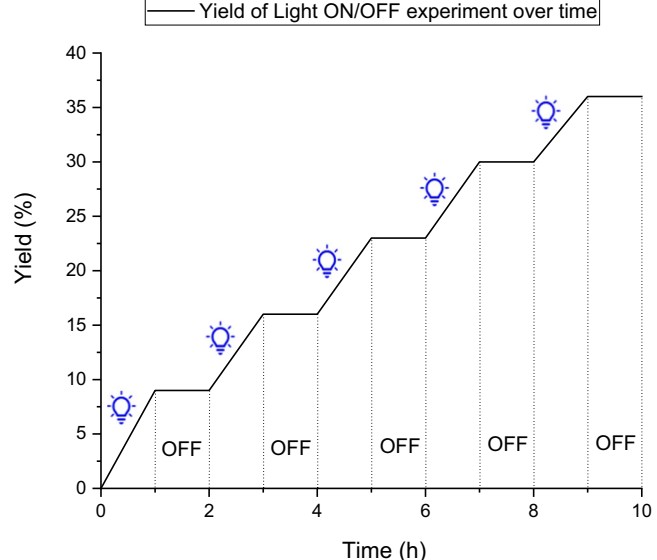

**Fig. 6 | Light on/off experiments.** Under the symbol of a light bulb corresponds to the period of light-induced reactions, above the term 'OFF' corresponds to dark reactions, and the vertical axis represents the reaction yield.

noteworthy that, apart from conventional phenyl-boronic acid, boronic esters could also serve as substrates, leading to the desired oxygenation products with good efficiency (**2p**), thereby expanding the range of substrates. However, unprotected amino groups posed a challenge in obtaining the desired products by this protocol. This was due to the electron-rich nature of amino groups, which competed with the donor and hindered the formation of the EDA complex. Nevertheless, this issue could be easily resolved by implementing a simple functional group protection strategy. In terms of alkyl substrates, we also attempted using secondary and tertiary alkylboron compounds as substrates, and achieved moderate to good yields of the target products.

Interestingly, the structurally complex boron-containing pharmaceutical molecule, bortezomib, worked well under the present EDA Complex photochemical oxidation-oxygenation reaction system (Fig. 5), showing its application in the late-stage modification of complex chemicals. Furthermore, to highlight the flexibility and application potential of this work, we also attempted using other types of substrates to capture the generated superoxide radicals. Interestingly, we have discovered that widely prevalent organic phosphorus

compounds and organosilicon compounds can also be transformed into oxygenated products (Fig. 5).

Stern-Volmer quenching experiment gives the constant $K_{SV} = 94.1$ (see Supplementary Figs. 14 and 15). As the concentration of the donor increases, the saturation of coordination leads to a decrease in the Stern-Volmer constant ($K_{SV}$) slope, which also evidences the existence of the EDA Complex. Furthermore, the results from Cyclic Voltammetry (CV) experiments also support the interaction of Doner and Acceptor (see Supplementary Figs. 17–20). To investigate the possibility of a chain reaction occurring within this system, we conducted tests and calculations for the quantum yield, ultimately determining that the quantum yield for the standard reaction is 2.39. Additionally, the results of the light on/off experiments are presented in Fig. 6. Combining the experimental results and related literature[67], we can conclude that the reaction is likely a light-driven chain reaction. The illumination provides the necessary energy for initiating and maintain the chain reaction. The experimental

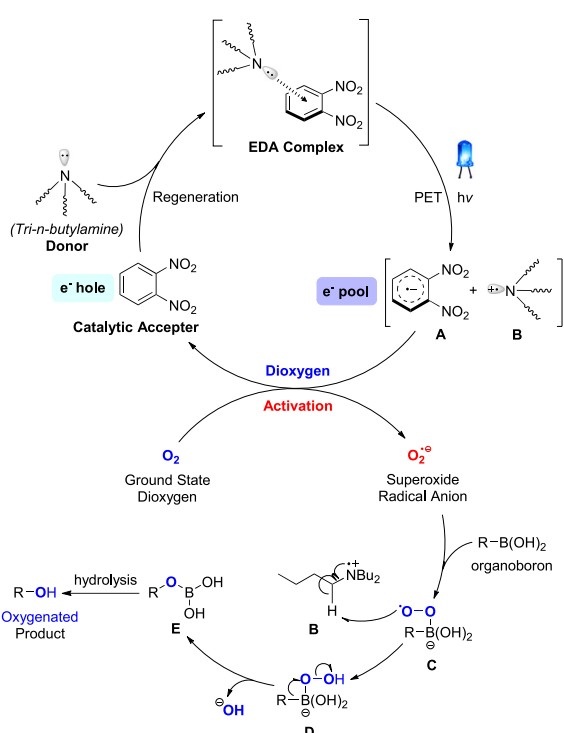

**Fig. 7 | Proposed EDA catalytic aerobic oxygenation mechanism.** The reaction mechanism is divided into two main parts: the upper half involves the catalytic cycle of oxygen activation leading to the formation of superoxide, while the lower half describes the oxygenation process of organic boron substrates.

findings, coupled with studies on the mechanism in related literature, hint at the existence of short chain reactions. Specifically, this involves the formation of oxidizing peroxide or radical intermediates through the reaction of certain amine radical intermediates with oxygen[68,69].

Based on the aforementioned experimental results and relevant mechanistic validation experiments, the reaction mechanism is proposed in Fig. 7. Initially, tri-*n*-butylamine as the electron donor interacts with a catalytic amount of 1,2-dinitrobenzene, which served as the electron acceptor in solution, leading to the formation of the EDA complex. This EDA complex has a visible light absorption charge-transfer band. Under excitation by visible light, a single electron transfer (SET) process takes place, where an electron from the donor is transferred to the acceptor, resulting in the formation of a diradical cage in acetonitrile solution. The electron-accepting intermediate, 1,2-dinitrobenzene radical anion **A**, is stabilized by the presence of two strongly electron-withdrawing nitro groups, preventing unproductive back electron transfer (BET). Subsequently, the dioxygen molecule captures the electron from the dinitrobenzene radical anion **A**, generating a superoxide radical anion, which is a kind of classical reactive oxygen species (ROS) with strong oxidizing capability. Simultaneously, the acceptor is regenerated, completing the catalytic cycle. The superoxide anion radical is then trapped by boronic acid, forming a peroxide radical intermediate **C**, which undergoes hydrogen atom transfer (HAT) with the generated cationic amine radical intermediate **B** to form the peroxyl alcohol species **D**. Subsequently, the rearrangement of intermediate **D** occurs to produce intermediate **E**, which is further hydrolyzed to yield the final oxygenation product.

In conclusion, through rational mechanism design, we have developed an EDA complex photocatalytic system that overcomes the oxygen-sensitive limitation of traditional EDA photocatalytic systems. This system innovatively achieves oxygen activation under mild conditions without the participation of transition metals, photoredox catalysts, and electrochemical equipment. The present dioxygen

activation and oxidation systems is mild, and highly compatible with different sensitive functional groups, such as thioethers, and aldehydes et. al. As a highly promising approach for dioxygen activation, this protocol is expected to have tremendous potential for a wide range of applications in oxygenation reactions.

## Methods

### General procedure for aerobic oxygenation of boric acid

To a dry 25 mL Schlenk tube, 0.5 mmol of boronic acid (if solid) and 4.3 mg (5 mol%) of 1,2-dinitrobenzene (1,2-DNB) were added. The tube was evacuated under reduced pressure and refilled with oxygen (3 times). Next, 120 μL of tri-n-butylamine (0.5 mmol, 1.0 eq) was added as a donor to the Schlenk tube. Anhydrous MeCN (4.0 mL) was then added using a syringe. The mixture was stirred at room temperature for 48 h under irradiation with blue LED (450 nm). After completion of the reaction, the solvent was removed under vacuum and the crude products were purified by column chromatography. For complete experimental details, including Photochemical instrumentation, related detection, procedures and full characterization ($^1$H and $^{13}$C NMR, HRMS spectrometry) of all new compounds, see Supplementary Information.

## Data availability

The authors declare that the data supporting the findings of this study are available within the paper or its Supplementary Information files and from the corresponding author upon request.

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

## Acknowledgements
The authors acknowledge the National Key R&D Program of China (No. 2021YFA1501700), the NSFC (Nos. 22293014, 22131002, 22161142019), Changping Laboratory, the New Cornerstone Science Foundation through the New Cornerstone Investigator Program and the XPLORER PRIZE for financial support. We thank Licheng Yang in this group for reproducing the results of **2c**, **2f** and **2r**.

## Author contributions
J.W., and N.J., conceived and designed the experiments; J.W. and J.M. carried out most of experiments; J.W., J.M., C.Z., Y. L., and N.J. analyzed data; J.W., Y. L. and N.J., wrote the paper; N.J. directed the project.

## Competing interests
The authors declare no competing interests.
