## [Peer Review File · Nature Communications]

Dioxygen Compatible Electron Donor-Acceptor Catalytic System and its Enabled Aerobic OxygenationREVIEWER COMMENTS

Reviewer #1 (Remarks to the Author):

In the present manuscript, the authors elegantly illustrate the use of the EDA complex as a catalytic system to produce the superoxide radical anion and subsequently execute a reaction replacing the boryl group with alcohol. The design to implement the EDA complex under O₂ conditions is an innovative approach. The detailed mechanism of superoxide radical anion formation via EDA complexation, alongside the design of the reaction mechanism employing a catalytic quantity of the acceptor in the EDA system, brings a distinctive dimension to this work. However, existing literature has highlighted that the superoxide radical anion can be easily synthesized using PC, and has been successfully applied to boronic acid prodrugs in cancer cells (JACS, 2023, 145, 18, 10082). In this context, the reaction in the manuscript, using the superoxide radical anion derived from the EDA complex to substitute the boryl group with alcohol, may seem like a familiar pathway. It might be pertinent to explore if using PC in catalytic proportions could offer similar or enhanced efficiencies. While the manuscript has its merits, there are aspects that could be further expanded upon. Given these considerations, I am hesitant to recommend the current version of the manuscript for publication in Nature Communications. However, I would be happy to re-evaluate a revised manuscript addressing these concerns.

-The exclusive utilization of borane to capture the superoxide radical somewhat narrows the study's breadth. Diversifying the applications to underscore the flexibility of other reaction systems via the superoxide radical anion capture could potentially elevate the work's significance.

-The manuscript mentions the reaction's sensitivity to air in a photoredox setting. It would be interesting in further insights regarding its behavior in general ambient air, and not limited to O₂ environments.

Reviewer #2 (Remarks to the Author):

In this submitted manuscript, Jiao and co-workers report a novel method to activate oxygen and generate superoxide radical anions under mild conditions without the presence of transition metals, photoredox catalysts or electrochemical equipment. Additionally, the authors also use boronic acids to react with this activated oxygen species to further demonstrate the utility of this process. The reaction exhibits great tolerance of functional groups, including alkyne, olefin, sulfide, or aldehyde. Both electron withdrawing and donating groups give good yields. Besides, the reaction also works with aromatic, alkyl boronic acids or even boronic esters.

The advancement of the paper lies within the design of the system to activate oxygen molecules. Traditional EDA complex system requires a stoichiometric amount of both donor and acceptors along with inert atmosphere. However, this manuscript only uses a stoichiometric amount of donor along with a catalyst amount of acceptor to activate oxygen molecule. This inspiration came from the work of König et. al., in which they used perylene diimides (PDI) along with triethylamine to act as an electron donor in the absence of oxygen to generate a stable radical anion. By substituting PDI with 1,2-dinitrobenzene, Jiao and co-workers are able to perform the similar process with the presence of oxygen to generate

reactive oxygen species (ROS). The authors also conduct a series of experiments to confirm the formation of ROS using DCFH and DHE fluorescence probe. The products are well characterized, and the manuscripts are written in an articulate and scientific manner. Thus, publication in Nat Comm is highly recommended, if the authors could address the following minor concerns:

1. Most of the substrate are aromatic boronic acids and alkyl boronic acids. There isn't any drug-like molecules. Can the authors perform some transformation using drug molecules or bioactive compounds? If boronic acids are uncommon in drug molecules or natural products, can the author perform a stream-line synthesis of a phenol derivative to demonstrate the utility of this method in pharmaceutical area?
2. In terms of alkyl boronic acids, only primary boronic acids are included in the substrate scope. Does the reaction also work with secondary or tertiary boronic acids?

Reviewer #3 (Remarks to the Author):

In this study, Jiao and coworkers have proposed a catalytic Electron Donor-Acceptor (EDA) comprising 1,2-DNB and tertiary amine that is compatible with dioxygen and facilitate aerobic oxygenation when exposed to visible light. It has been proposed that 1,2-DNB serves as a catalytic acceptor and it's EDA complex with amine oxidizes both aromatic and aliphatic boronic acids into their respective alcohols under the visible light irradiation. To substantiate the findings, the authors conducted ultraviolet-visible (UV-Vis) spectroscopy tests along with other control experiments. The UV-Vis experiments indicated that the acceptor molecule exhibits absorption in the visible region, with the charge transfer (CT) band also observed within the same region. Although a slight bathochromic shift is present, both absorptions occur only within the visible range. This phenomenon gives rise to a state of uncertainty regarding the underlying mechanism involved. A light source should be selected that possesses an optical profile that does not coincide with any local band. It is highly likely that the mechanism involves a photoredox catalytic cycle, where 1,2-DNB serves as a photocatalyst and the base as a reductive quencher. This highly reducing catalyst then reduces dioxygen to the corresponding radical anion species. Authors should have performed more control experiments to justify their claim. A simple Stern-Volmer analysis would have been beneficial. The detailed analysis involving photophysical studies and CV is missing. Furthermore, quantum yield calculation and light on/off experiments should be performed to probe the existence of a radical chain mechanism. The authors should also perform other type of reactions to demonstrate the generality of the concept. In general, the work is not detailed and comprehensive and lacks novelty. Therefore, the reviewer does not recommend this work to be published in Nature Communications.

We appreciate reviewers very much for reviewing our manuscript NCOMMS-23-40237. Here I would answer all the comments from reviewers **point by point** and can make revisions as following:

Reviewer #1: Comments

In the present manuscript, the authors elegantly illustrate the use of the EDA complex as a catalytic system to produce the superoxide radical anion and subsequently execute a reaction replacing the boryl group with alcohol. The design to implement the EDA complex under O₂ conditions is an innovative approach. The detailed mechanism of superoxide radical anion formation via EDA complexation, alongside the design of the reaction mechanism employing a catalytic quantity of the acceptor in the EDA system, brings a distinctive dimension to this work. However, existing literature has highlighted that the superoxide radical anion can be easily synthesized using PC, and has been successfully applied to boronic acid prodrugs in cancer cells (JACS, 2023, 145, 18, 10082). In this context, the reaction in the manuscript, using the superoxide radical anion derived from the EDA complex to substitute the boryl group with alcohol, may seem like a familiar pathway. It might be pertinent to explore if using PC in catalytic proportions could offer similar or enhanced efficiencies. While the manuscript has its merits, there are aspects that could be further expanded upon. Given these considerations, I am hesitant to recommend the current version of the manuscript for publication in Nature Communications. However, I would be happy to re-evaluate a revised manuscript addressing these concerns.

Answer: Thank the reviewer very much for reviewing our manuscript. The positive view and valuable questions and suggestions are all highly appreciated. The manuscript has been carefully revised based on the valuable comments.

As the reviewer mentioned, there indeed are reports in the literatures about obtaining phenolic compounds through oxidation of phenylboronic acid with dioxygen. However, our research focus is significantly different. Represented by the literature cited by the reviewer (JACS, 2023, 145, 18, 10082), their main innovation lies in utilizing the photo-oxidation-reduction properties of traditional photocatalysts to generate reactive oxygen species that oxidize the substrate. As discussed in the reported literatures, the activation of oxygen through photochemical strategies requires an iridium (III) complex or organic dye methylene blue (MB) as the photocatalyst for dioxygen activation.

In contrast, without using traditional photocatalysts, this manuscript highlights our recent findings on the photochemical properties of Electron Donor-Acceptor (EDA) complexes and their potential applications in synthetic chemistry. In particular, we address the inherent limitation of oxygen sensitivity associated with traditional EDA photocatalytic systems. Through rational design, we have developed a novel

EDA complex photocatalytic system that overcomes the oxygen-sensitive limitation of traditional EDA photocatalysis, and provides a new protocol for producing the superoxide radical through dioxygen activation:

- 1) Our research is an exploration on the application of EDA complex in photochemistry. Recently, electron donor-acceptor (EDA) complex photochemistry has garnered increased attention, which did not require any additional photocatalyst for successful execution and provided new opportunities for synthetic chemistry (Scheme A1). Despite great significance, these reports are oxygen-sensitive and limited to the oxidant-free transformations. By rational design, we developed a novel EDA complex to address this problem (Scheme A2).

Scheme A1. Recent works that involve EDA complex photochemistry.

Scheme A2. This work: A oxygen compatible EDA complex and its enabled oxygen

activation.

- 2) The present chemistry provides a new protocol for producing the superoxide radical via dioxygen activation. Considering that the importance of dioxygen activation in the sustainable aerobic oxidation/oxygenation reactions, we envisioned that an EDA complex photochemistry strategy via two-steps electron relay would be able to realize O₂ activation and superoxide generation.
- 3) We provide an organocatalytic acceptors for EDA Complex photoactivation. While the advancement of EDA complex photochemistry has alleviated the need for photocatalysts in photochemical transformations, it is important to note that the most of reported work in this field still rely on the participation of stoichiometric donor and acceptor substrates. Recently, there have been successful implementations of EDA complex photochemistry within catalytic regimes, utilizing catalytic donors (Scheme A3a). However, organocatalytic acceptors in this field are still rare, these was one case that requires the assistance of redox auxiliaries (Scheme A3b). Our study provides a comprehensive elucidation of how electron-deficient nitroaromatic compounds can function as organocatalytic acceptors, effectively facilitating the entire EDA complex photochemical transformation process (Scheme A3c).

Science, **2019**, 363, 1429.

J. Am. Chem. Soc. **2021**, 143, 12304.

Complex with **catalytic acceptor** with the help of redox auxiliaries (**RA**):

J. Am. Chem. Soc. **2022**, 144, 8914.

Work: EDA complex with **catalytic acceptor**

Scheme A3. Catalytic donors and acceptors in EDA complex photochemistry.

Additionally, we have conducted deeply mechanistic studies, monitored the interaction between electron donor and acceptor, and demonstrate the presence of superoxide radicals. We discovered that the presence of the nitro group in the acceptor serves as a crucial stabilizing factor for the radical anionic intermediate, thereby significantly suppressing the occurrence of unproductive back electron transfer (BET). These findings highlight the remarkable potential of nitroaromatic compounds as efficient and reliable acceptors in EDA complex photochemistry. We believe that this research offers an innovative approach for the advancement of fields including oxygen activation, EDA Complex, and photochemistry.

1. The exclusive utilization of borane to capture the superoxide radical somewhat narrows the study's breadth. Diversifying the applications to underscore the flexibility of other reaction systems via the superoxide radical anion capture could potentially elevate the work's significance.

Answer: We appreciate the reviewer's kind suggestion. We have investigated the oxygenation of phosphorus heteroatoms and silicon compounds. Interestingly, we have discovered that widely prevalent organic phosphorus compounds and organosilicon compounds can also be transformed into oxygenated products (Scheme A4). These results have been added in the revised manuscript.

Scheme A4. Diversifying the applications by oxygenation of other type of substrates.

2. The manuscript mentions the reaction's sensitivity to air in a photoredox setting. It would be interesting in further insights regarding its behavior in general ambient air, and not limited to O₂ environments.

Answer: The reviewer's suggestion is gratefully appreciated and is indeed very meaningful and of practical value. Under standard conditions, by simply changing the oxygen atmosphere to **ambient air**, the reaction proceeded smoothly and yielded the target product but with a little bit lower efficiency (Scheme A5). This result has been added in revised manuscript at Table 1.

Scheme A5. Oxygenation reactivity in ambient air.

Reviewer #2: Comments

In this submitted manuscript, Jiao and co-workers report a novel method to activate oxygen and generate superoxide radical anions under mild conditions without the presence of transition metals, photoredox catalysts or electrochemical equipment. Additionally, the authors also use boronic acids to react with this activated oxygen species to further demonstrate the utility of this process. The reaction exhibits great tolerance of functional groups, including alkyne, olefin, sulfide, or aldehyde. Both electron withdrawing and donating groups give good yields. Besides, the reaction also works with aromatic, alkyl boronic acids or even boronic esters.

The advancement of the paper lies within the design of the system to activate oxygen molecules. Traditional EDA complex system requires a stoichiometric amount of both donor and acceptors along with inert atmosphere. However, this manuscript only uses a stoichiometric amount of donor along with a catalyst amount of acceptor to activate oxygen molecule. This inspiration came from the work of König et. al., in which they used perylene diimides (PDI) along with triethylamine to act as an electron donor in the absence of oxygen to generate a stable radical anion. By substituting PDI with 1,2-dinitrobenzene, Jiao and co-workers are able to perform the similar process with the presence of oxygen to generate reactive oxygen species (ROS). The authors also conduct a series of experiments to confirm the formation of ROS using DCFH and DHE fluorescence probe. The products are well characterized, and the manuscripts are written in an articulate and scientific manner.

Thus, publication in Nat Comm is highly recommended, if the authors could address the following minor concerns:

Answer: Thank the reviewer so much for reviewing our manuscript. The positive view and valuable questions and suggestions are all highly appreciated. The manuscript has been carefully revised based on the valuable comments.

1. Most of the substrate are aromatic boronic acids and alkyl boronic acids. There isn't any drug-like molecules. Can the authors perform some transformation using drug molecules or bioactive compounds? If boronic acids are uncommon in drug molecules or natural products, can the author perform a stream-line synthesis of a phenol derivative to demonstrate the utility of this method in pharmaceutical area?

Answer: Thank the reviewer so much for the kind suggestion. We experimented with the structurally complex boron-containing pharmaceutical molecule, bortezomib, as our substrate. Employing the EDA Complex photochemical oxidation-oxygenation reaction system developed by us, we successfully modified it into an oxygenated derivative in the late stages (Scheme A6). This result has been added in revised manuscript.

Scheme A6. Late-stage modification of drug molecule.

2. In terms of alkyl boronic acids, only primary boronic acids are included in the substrate scope. Does the reaction also work with secondary or tertiary boronic acids?

Answer: Thank the reviewer very much for the constructive question. We attempted using secondary carbon alkylboronic acids as substrates, such as cyclohexylboronic acid, and excitingly, the secondary carbon boronic acid substrates were able to yield the target product well. Despite the limited availability of commercially obtainable tertiary carbon boronates, we still tried using tert-butyl boronate as the substrate. Even though there was significant steric hindrance, we were still able to satisfactorily obtain the target product (Scheme A7). *Note:* Since the products are low molecular weight alcohols like cyclohexanol and *t*-butanol, their volatile nature and low boiling points make it difficult to separate and purify. Therefore, we monitored the reaction process and the target product using NMR with 1,1,2,2-Tetrachloroethane (TCE) as internal standard.

Scheme A7. Oxygenation with secondary or tertiary boronic acids.

Reviewer #3: Comments

In this study, Jiao and coworkers have proposed a catalytic Electron Donor-Acceptor (EDA) comprising 1,2-DNB and tertiary amine that is compatible with dioxygen and facilitate aerobic oxygenation when exposed to visible light. It has been proposed that 1,2-DNB serves as a catalytic acceptor and its EDA complex with amine oxidizes both aromatic and aliphatic boronic acids into their respective alcohols under the visible light irradiation. To substantiate the findings, the authors conducted ultraviolet-visible (UV-Vis) spectroscopy tests along with other control experiments.

The UV-Vis experiments indicated that the acceptor molecule exhibits absorption in the visible region, with the charge transfer (CT) band also observed within the same region. Although a slight bathochromic shift is present, both absorptions occur only within the visible range. This phenomenon gives rise to a state of uncertainty regarding the underlying mechanism involved. A light source should be selected that possesses an optical profile that does not coincide with any local band.

Answer: Thank the reviewer very much for reviewing our manuscript and the valuable question. Indeed, as the reviewer mentioned, UV-Visible absorption spectroscopy tests showed that the acceptor catalyst 1,2-dinitrobenzene used under our optimal conditions absorbs in the visible light region. However, after the formation of the EDA Complex by the Donor and Acceptor, a significant and visually observable change in absorption occurred (Figure 1 in the manuscript), which is a characteristic feature of EDA Complex formation and cannot be observed in conventional photoredox catalysts systems (like transition metal complexes, conjugated organic dyes, etc.).

Furthermore, according to the reviewer's suggestion, in our system, it is also feasible to use a light source that possesses an optical profile that does not coincide with any local band. For example, 1,3-dinitrobenzene (1,3-DNB) has almost no absorption in the visible light region, especially no absorbency near the wavelength of *450 nm* of our light source used. However, upon adding the Donor, the formation of the EDA complex results in the emergence of new absorption bands (visible absorption, especially around 450 nm), please see **Fig. A1**. The absorbance of these new bands increases with the addition of the Donor (reflecting the dynamic equilibrium of Donor-Acceptor binding, and the addition of 20 times the amount of Donor to the Acceptor used as a catalyst also corresponds to the actual usage ratio of the relevant compounds: 5 mol% vs 1.0 eq). It was further supported that the reaction with 1,3-dinitrobenzene instead of 1,2-dinitrobenzene as the Acceptor produced the corresponding phenol product in 61% (Scheme A8). These results demonstrate the rationality of the EDA Complex photocatalytic system we proposed, which have been added in the revised manuscript and SI.

Figure A1. 1,3-DNB and the corresponding absorption bands of EDA complex

Scheme A8. The efficiency of the EDA complex catalytic reaction corresponding to 1,3-DNB.

It is highly likely that the mechanism involves a photoredox catalytic cycle, where 1,2-DNB serves as a photocatalyst and the base as a reductive quencher. This highly reducing catalyst then reduces dioxygen to the corresponding radical anion species. Authors should have performed more control experiments to justify their claim.

Answer: We express gratitude for the reviewer's evaluation. As they mentioned, in *conventional photoredox systems* (like transition metal complexes, conjugated

organic dyes, etc.), the photocatalyst, upon absorbing photons and being excited, undergoes reductive quenching through single-electron transfer (SET) with variety of amine sacrificial agents acting as reductive quenchers. Meanwhile, the reduced state of the photocatalyst, due to its strong reducing ability, will transfer electrons to oxygen. However, the present EDA complex photocatalytic system we have demonstrated is fundamentally different from traditional photoredox catalytic systems based on the reasons as follows:

Firstly, nitrobenzene compounds are not photoredox catalysts in the traditional sense. The photophysics of nitroaromatic compounds is characterized by an ultrafast decay into the triplet manifold and by significant triplet quantum yields (*Phys. Chem. Chem. Phys.* **2019**, *21*, 10514). The photochemical process involving nitrobenzene compounds typically require ultraviolet light excitation, through $n\text{-}\pi^*$ transitions to the singlet excited state, and then rapidly undergoing intersystem crossing (ISC) to the triplet state (*J. Am. Chem. Soc.* **1966**, *88*, 4330.). Therefore, single-electron transfer (SET) occurring between singlet excited state nitroaromatics and amine sacrificial agents is unlikely.

Secondly, in the triplet excited state, nitroaromatics as highly active biradicals can react with a variety of organic compounds, including solvents. For instance, they can undergo hydrogen atom transfer reactions with reactive carbon-hydrogen bonds (such as alcohols), oxygen addition and then carbon-carbon bond cleavage reactions with olefins, and so on (*J. Am. Chem. Soc.* **1966**, *88*, 4330; *Nature*, **2022**, *610*, 81; *J. Am. Chem. Soc.* **2022**, *144*, 15437.). However, in the EDA Complex photocatalytic system we developed, alcohols as oxygenated desired products of alkylboron compounds are obtained with a satisfactorily high yield. The system shows excellent compatibility with a variety of functional groups. Especially for the olefins mentioned earlier, which can quickly react with excited state nitroaromatics, no by-products of oxygen addition or carbon-carbon bond cleavage were observed. The aforementioned results exclude the possibility of the existence of excited-state nitroaromatics (DNB).

Thirdly, as for amines used as sacrificial reagents in typical photoredox catalytic systems, their type (steric hindrance) generally has little impact on the yield. This is because the excited state photocatalyst has very high activity, and its redox ability is greatly enhanced due to the rearrangement of electrons in the molecular orbitals in the excited state. Therefore, various types of amines and even other types of reducing agents can serve as sacrificial reducing agents as long as the redox potentials are mutually compatible (*J. Org. Chem.* **2016**, *81*, 6898; *Angew. Chem. Int. Ed.* **2020**, *59*, 17356; *Chem. Rev.* **2016**, *116*, 10075; *Chem. Rev.* **2013**, *113*, 5322.). However, in the EDA Complex photocatalytic system, due to the electron-rich amine compounds acting as Donors, the process of forming EDA complexes with electron-deficient acceptor is extremely sensitive to steric hindrance: The reaction yields corresponding to donors with similar redox potentials (*Org. Lett.* **2020**, *22*, 2822; *Synlett*, **2016**, *27*, 714.) but differing steric hindrances vary significantly. (See **Scheme A9** and **Table S1**.)

Optimization of Electron Donor in SI). The yield of the reaction is sensitive to the steric hindrance of the amine rather than to the redox potential of the amine, further corroborating the mechanism of the EDA Complex photocatalytic system we proposed.

(*Tips:* There are numerous literature reports on amine compounds acting as reductive quenchers, such as: THIQ: *Org. Lett.* **2012**, *14*, 672; Secondary amines: *Angew. Chem. Int. Ed.* **2012**, *51*, 222. Et₃N: *Nat. Commun.* **2021**, *12*, 478; n-Bu₃N: *Angew. Chem., Int. Ed.* **2011**, *50*, 9655; TPA: *New J. Chem.* **2020**, *44*, 19061; DIPEA: *J. Am. Chem. Soc.* **2009**, *131*, 8756.)

Scheme A9. Yields corresponding to donors with similar redox potentials but different steric hindrances.

In conclusion, the dioxygen activation system we have developed is not only distinct from traditional photoredox catalytic systems but also overcomes various limitations of conventional EDA complex systems, making it an innovative methodology both mechanistically and conceptually.

The above results have been added in the revised SI.

A simple Stern-Volmer analysis would have been beneficial. The detailed analysis involving photophysical studies and CV is missing.

Answer: We are grateful for the valuable suggestion provided by the reviewer. To enhance the importance of this research, we conducted a Stern-Volmer analysis, and the results are as follows (**Figure A2 – A3**):

Figure A2. Stern-Volmer quenching experiment

Figure A3. Stern-Volmer quenching experiment linear fit

Figure A4. The fluorescence quenching curve with extended concentration.

It is noteworthy that as the amount of the donor increases, the fluorescence quenching curve no longer remains linear. This is related to the coordination saturation of the EDA Complex: with the addition of the donor, EDA Complexes are continuously formed, rapidly quenching the luminescence of the acceptor. However, as the amount of the donor increases further, the formation of the EDA Complex gradually reaches saturation, resulting in a decrease and eventual leveling off of the slope of the quenching curve (**Figure A4**). This observation aligns with the mechanism proposed in our study.

The results of the Cyclic Voltammetry (CV) experiment are as follows (**Figure A5 – A8**):

Figure A5. CV of Acceptor 1,2-DNB

Figure A6. CV of Donor TBA

Figure A7. CV of EDA Complex

Figure A8. Merged CV of EDA Complex together with Donor and Acceptor

Analysis and discussion of the CV experimental results: The redox potential of the EDA Complex shifted to the left (towards a lower potential) compared to the isolated Donor and Acceptor (**Figure A8**). This shift is attributed to the formation of the EDA Complex, where the electron clouds of the Donor and Acceptor molecules

become polarized due to their interaction. This polarization causes an elevation in the energy level of the Donor's HOMO (Highest Occupied Molecular Orbital), making it more easily oxidizable. Concurrently, once the Acceptor molecule combines with the electron-rich Donor, the resultant EDA Complex is more inclined towards intracomplex electron transfer, thus becoming less receptive to the reduction by external electrons.

Note: All electrode potentials have been calibrated using the ferrocene standard potential and all data were measured three times and the averages were taken to ensure accuracy.

The above results have been added in the revised SI.

Furthermore, quantum yield calculation and light on/off experiments should be performed to probe the existence of a radical chain mechanism.

Answer: We sincerely appreciate the insightful recommendation offered by the reviewer. To investigate the possibility of a chain reaction occurring within this system, we conducted relevant experiments as follows:

$$\text{mol Fe}^{2+} = \frac{V \times \Delta A}{l \times \epsilon} = \frac{0.00235 \text{ L} \times 0.2275339}{1.000 \text{ cm} \times 11100 \text{ mol}^{-1} \text{cm}^{-1}} = 4.82 \times 10^{-8} \text{ mol}$$

Where V is the volume of the respective sample solution analyzed (2.35 mL), ΔA is the difference between the average absorbance of irradiated and nonirradiated ferrioxalate solutions at 510 nm, l is the pathlength, and the ϵ is the molar absorptivity at 510 nm.

$$\text{photon flux} = \frac{\text{mol Fe}^{2+}}{\phi \times t \times f} = \frac{4.82 \times 10^{-8} \text{ mol}}{0.92 \times 10 \text{ s} \times 0.99066} = 5.28 \times 10^{-9} \text{ einstein s}^{-1}$$

Where ϕ is the quantum yield for the ferrioxalate actinometer at 450 nm, t is the irradiation time and f are the fraction of light absorbed by the ferrioxalate actinometer solution.

$$f = 1 - 10^{-A} = 1 - 10^{-2.029595} = 0.99065875$$

Where A is the measured absorbance of the ferrioxalate actinometer solution at 456 nm before blue LED irradiation without 1,10-phenanthroline.

Determination of the fraction of light absorbed at 450 nm by the ferrioxalate solution: The absorbance of the ferrioxalate actinometer solution at 450 nm before blue LED irradiation, without 1,10-phenanthroline, was measured at 2.029595.

The quantum yield (Φ) has been calculated using the equation:

$$\Phi = \frac{\text{mol Product}}{f \times t \times \text{flux}} = \frac{0.5 \times 0.09 \times 10^{-3} \text{ mol}}{0.99065875 \times 3600 \text{ s} \times 5.28 \times 10^{-9} \text{ einstein s}^{-1}} = 2.39$$

Where t is the reaction time, f is previously calculated fraction of light absorbed by

the respective solution.

The results of the light on/off experiments are as follows:

Figure A9. light on/off experiments

Annotation: For the accuracy and reliability of the experimental results, all experimental data were repeated three times and the average values were taken.

Combining the experimental results and related literature (*Chem. Sci.* **2015**, *6*, 5426; *Org. Lett.* **2023**, *25*, 7863.), we can conclude that the reaction is likely a light-driven chain reaction. The illumination provides the necessary energy for initiating and maintain the chain reaction. The experimental findings, coupled with studies on the mechanism in related literature (*Beilstein J. Org. Chem.* **2013**, *9*, 1977; *Org. Biomol. Chem.* **2022**, *20*, 9503.), hint at the existence of short chain reactions. Specifically, this involves the formation of oxidizing peroxide or radical intermediates through the reaction of certain amine radical intermediates with oxygen.

The authors should also perform other type of reactions to demonstrate the generality of the concept.

Answer: We express our sincere gratitude for the professional and constructive suggestions provided by the reviewer. We have investigated the oxygenation of phosphorus heteroatoms and silicon compounds. Interestingly, we have discovered that widely prevalent organic phosphorus compounds and organosilicon compounds can also be transformed into oxygenated products (**Scheme A10**):

Scheme A10. Diversifying the applications by oxygenation of other type of substrates.

These results have been added in the revised manuscript.

In general, the work is not detailed and comprehensive and lacks novelty. Therefore, the reviewer does not recommend this work to be published in Nature Communications.

Answer: Thank the reviewer very much for reviewing our manuscript and the valuable suggestions, which are very helpful for us to improve our manuscript. It is perhaps due to unclear manuscript writing, the innovative aspects of this work were not sufficiently emphasized. Actually, without using traditional photocatalysts, this manuscript highlights our recent findings on the photochemical properties of Electron Donor-Acceptor (EDA) complexes and their potential applications in synthetic chemistry. In particular, we address the inherent limitation of oxygen sensitivity associated with traditional EDA photocatalytic systems. Through rational design, we have developed a novel EDA complex photocatalytic system that overcomes the oxygen-sensitive limitation of traditional EDA photocatalysis, and provides a new protocol for producing the superoxide radical through dioxygen activation:

- 1) Our research is an exploration on the application of EDA complex in photochemistry. Recently, electron donor-acceptor (EDA) complex photochemistry has garnered increased attention, which did not require any additional photocatalyst for successful execution and provided new opportunities for synthetic chemistry (Scheme A11). Despite great significance, these reports are oxygen-sensitive and limited to the oxidant-free transformations. By rational design, we developed a novel EDA complex to address this problem (Scheme A12).

Nature Chem., **2013**, 5, 750.

Science, **2017**, 357, 283.

Nature, **2020**, 584, 69.

Nature Chem., **2023**, 15, 43.

Scheme A11. Recent works that involve EDA complex photochemistry.

Scheme A12. This work: A oxygen compatible EDA complex and its enabled oxygen activation.

- 2) The present chemistry provides a new protocol for producing the superoxide radical via dioxygen activation. Considering that the importance of dioxygen activation in the sustainable aerobic oxidation/oxygenation reactions, we envisioned that an EDA complex photochemistry strategy via two-steps electron relay would be able to realize O_2 activation and superoxide generation.
- 3) We provide an organocatalytic acceptors for EDA Complex photoactivation. While the advancement of EDA complex photochemistry has alleviated the need for photocatalysts in photochemical transformations, it is important to note that the most of reported work in this field still rely on the participation of

stoichiometric donor and acceptor substrates. Recently, there have been successful implementations of EDA complex photochemistry within catalytic regimes, utilizing catalytic donors (Scheme A13a). However, organocatalytic acceptors in this field are still rare, these was one case that requires the assistance of redox auxiliaries (Scheme A13b). Our study provides a comprehensive elucidation of how electron-deficient nitroaromatic compounds can function as organocatalytic acceptors, effectively facilitating the entire EDA complex photochemical transformation process (Scheme A13c).

Science. **2019**, 363, 1429.

J. Am. Chem. Soc. **2021**, 143, 12304.

Complex with **catalytic acceptor** with the help of redox auxiliaries (**RA**):

J. Am. Chem. Soc. **2022**, 144, 8914.

ork: EDA complex with **catalytic acceptor**

Scheme A13. Catalytic donors and acceptors in EDA complex photochemistry.

Additionally, we have conducted deeply mechanistic studies, monitored the interaction between electron donor and acceptor, and demonstrate the presence of superoxide radicals. We discovered that the presence of the nitro group in the acceptor serves as a crucial stabilizing factor for the radical anionic intermediate, thereby significantly suppressing the occurrence of unproductive back electron transfer (BET). These findings highlight the remarkable potential of nitroaromatic compounds as efficient and reliable acceptors in EDA complex photochemistry. We believe that this research offers an innovative approach for the advancement of fields including oxygen activation, EDA Complex, and photochemistry.

Again: Thank all the reviewers very much. I hope this revised manuscript publishable in *Nature Communications*.

Best regards.

Sincerely,

Ning Jiao

Professor of Chemistry

State Key Laboratory of Natural and Biomimetic Drugs

Peking University,

Beijing 100083, China

Tel. & Fax: 86-10-8280-5297

E-mail: jiaoning@bjmu.edu.cn

REVIEWERS' COMMENTS

Reviewer #1 (Remarks to the Author):

I appreciate the authors' efforts in responding to questions and enhancing the manuscript. The incorporation of the EDA system in the reaction mechanism introduces a unique aspect to this study. However, I realized that there are already many efficient methods for converting boronic acids into hydroxy groups using a visible-light-mediated photoredox strategy involving the superoxide radical anion (Angew. Chem. Int. Ed. 2012, 51, 784, JACS 2013, 135, 13286, Synlett 2014, 2613, and RSC Adv. 2015, 5, 65016504...). These existing systems, employing PC such as methylene blue and an amine base, are fundamentally similar to the reaction components and conditions of the current study, except for the EDA system. Moreover, the Itoh group has demonstrated that such transformations can be achieved even without a base, further simplifying the process. In the present study, the extended reaction time challenges the assertion of technical advancement. Considering the simplicity of the reaction, which involves converting boron into a hydroxy group using the superoxide radical anion, I believe this work falls short of the novelty standards set by Nat. Commun. Therefore, I recommend transferring this manuscript to a more specialised journal.

Reviewer #2 (Remarks to the Author):

The revised manuscript fully addressed our concerns. The additional mechanistic experiments is thoroughly studied and improved the quality of this manuscript. Therefore, publication in Nat Comm is recommended.

Reviewer #3 (Remarks to the Author):

The revised manuscript is substantially improved and the reviewer is satisfied with the rebuttal letter. The authors have put noteworthy effort to address this and other reviewer's queries/suggestions. Hence, the manuscript is recommended to be published in the present form.